# An Assessment of Training Characteristics Associated with Atrial Fibrillation in Masters Runners

**DOI:** 10.3390/sports7070179

**Published:** 2019-07-23

**Authors:** Martin E. Matsumura, Justin R. Abbatemarco

**Affiliations:** 1Geisinger Health System, Pearsall Heart Hospital, Wilkes Barre, PA 18711, USA; 2Mellen Center for Multiple Sclerosis Treatment and Research, Cleveland Clinic, Cleveland, OH 44195, USA

**Keywords:** atrial fibrillation, running, exercise

## Abstract

A growing body of literature supports an association between long-term endurance exercise and the development of atrial fibrillation (AF). Given the benefits of lifelong exercise, a better understanding of this association is critical to allow healthcare providers to counsel aging exercisers on the proper “dose” of exercise to maximize health benefits but minimize AF risk. The current study examines the relationship between specific aspects of training volume and intensity and the occurrence of AF among older runners in order to better understand what aspects of endurance exercise may contribute to the development of AF. The study was an Internet-based survey of endurance training and health characteristics of runners 35 years of age and older. A total 2819 runners participated and 69 (2.4%) reported a current or prior diagnosis of AF. Among “traditional” risk factors, runners reporting AF were older, more likely to be male, and had higher rates of hypertension and diabetes. Among training characteristics, only accumulated years of training was associated with AF. In contrast, average weekly mileage, training pace, and days of training per week were not associated with AF. In a multivariable analysis that included chronologic age, sex, diabetes, and hypertension, accumulated years of training remained significantly associated with the report of AF. These findings suggest that the relationship between chronic endurance exercise and AF is dependent on the accumulated training duration but does not appear to be influenced by specific training characteristics such as frequency or intensity of endurance exercise. Further confirmation of these relationships may help healthcare providers counsel exercisers on optimal training habits and identify endurance athletes who are at risk for the development of AF.

## 1. Introduction

The benefits of exercise on cardiovascular health are supported by a wealth of data [1,2,3], and as a result, there is growing emphasis on the promotion of lifelong participation in exercise and an active lifestyle to maintain optimal health with aging. While the cardiovascular benefits of exercises are unequivocal, accumulating evidence is suggestive of an association between chronic participation in endurance exercise, such as distance running, and the development of atrial fibrillation (AF) [4,5,6,7,8]. The mechanisms of this association are not well understood but are hypothesized to include left atrial fibrosis and remodeling, systemic inflammation, and increased vagal tone [9,10]. The concept of AF promoted by high volume and/or high intensity exercise is countered by recent studies supporting the use of more moderate amounts of exercise training to reduce the incidence and burden of AF [11]. Thus, while it appears that there is likely an optimal amount and intensity of exercise beyond which the development of AF becomes more common, what remains unclear is whether there are specific characteristics of high-level training which result in the promotion of AF in endurance exercisers [12]. The present study examined self-reported history of AF and training habits of chronic habitual runners in order to define whether there are specific training variables that are independently associated with the occurrence of AF.

## 2. Methods

The Masters Athletic Study is a longitudinal, self-reported Web-based study of training practices and health characteristics in runners ages 35 years and older. The study was launched from a HIPAA-compliant website in July 2014 and collected data from runners for a period running from July–August 2013. The details of the study have been described elsewhere [13]. Any runner who responded to the survey and was 18 years of age or older was invited to participate. For the present study, all participants who responded to the question regarding history of atrial fibrillation were included. Patient datasets were excluded if they reported an age less than 35 years. Participants were asked for detailed information regarding training duration (years of participation), volume (weekly mileage), intensity (average training pace), and frequency (average training days/week). Additionally, they were asked about specific characteristics known to be associated with AF (age, sex, diabetes, hypertension). Some traditional risk factors were not assessed in the survey and thus were not able to be measured in this study (obesity, alcohol use, smoking) [14,15]. Questions regarding training characteristics were open-ended and did not specify a specific time range (e.g., years of running was queried as “For how many years have you been running?”). For questions regarding running characteristics, answers to questions were answered in intervals (years of training 0–10 years, 11–20 years, etc.). Demographic variables were answered as either “yes/no” (diabetes, HTN, etc.) or a discreet number (age).

Respondents were grouped based on a self-reported diagnosis of atrial fibrillation and were omitted from the current analysis if they did not complete the questionnaire or reported a history of heart disease including coronary artery disease, myocardial infarction, or heart failure.

Chi square analysis was used to compare the relationship of traditional risk factors to the report of AF and to compare the distribution of responses to training variables to the report of AF. Multivariable logistic regression analysis was used to assess which variables determined to be univariate predictors of AF were independent predictors. For all comparisons, *p* < 0.05 was considered significant. The study was approved by the institutional review board of Geisinger Health System. 

## 3. Results

For the present study, survey responses were received from 3517 participants. Of the 3517 respondents, 2819 (80.1%) provided complete answers regarding history of AF and training history and were included in the assessment. The mean age of the respondents was 49.1 years (range 35–90) and 68.2% of the respondents were male. Disease history was as follows: hypertension 20.9%, hyperlipidemia 21.9%, diabetes mellitus 1.4%, family history of cardiovascular disease 40.7%.

A current or prior diagnosis of AF was reported by 69 (2.4%) respondents. The prevalence of traditional risk factors for AF was compared between those reporting AF vs. those who did not report AF. Significant differences were found in mean age (59.8 vs. 48.4 years, *p* < 0.001), male sex (85.5% vs. 67.8%, *p* = 0.003), and report of hypertension (47.8% vs. 20.2%, *p* < 0.001) and diabetes mellitus (7.2% vs. 1.3%, *p* < 0.001) between those with vs. without AF, respectively. 

Training habits were compared between the two groups and are depicted in Figure 1.

Of all of the training characteristics assessed, only the distribution of accumulated years of training was significantly different between those participants reporting AF vs. those not reporting AF (Figure 1A, χ^2^ = 37.0, *p* < 0.001). There was a clear relationship between years of accumulated training and AF, with AF reported by 1.2% of runners with up to 10 years accumulated training, 1.3% of runners with 11–20 years, 2.4% of runners with 21–30 years, and 6.1% of runners with >30 years of training (data not shown). No other training characteristic was associated with AF, including average training pace (Figure 1B, χ^2^ = 4.4, *p* = 0.35), average weekly mileage (Figure 1C, χ^2^ = 5.6, *p* = 0.13), and average days of training per week (Figure 1D, χ^2^ = 3.2, *p* = 0.20).

To confirm that the relationship between accumulated years of training and AF is not simply due to the direct relationship between accumulated years of training and older chronologic age in the cohort reporting AF, we constructed a multivariable model containing both chronologic age and years of training as well as sex, hypertension, and diabetes (variables associated with AF in univariate analysis, Table 1).

## 4. Discussion

The benefits of lifelong exercise to promote cardiovascular health are supported by robust data, and the recently released ACC/AHA prevention guidelines continue to promote threshold levels of moderate to vigorous intensity physical activity to maintain cardiovascular health with aging [16]. While there is little downside to long-term, high-level endurance exercise, the development of AF has emerged as a rare “side effect” of this endeavor, with studies suggesting a rate between 6–10% [1,2,3]. The present study is the first to focus on runners aged 35 and above while examining the relationship of specific lifelong training regimens and variability in running volume and intensity with AF. We found that among a variety of training habits, the report of AF in our cohort was related only to years of accumulated endurance training. This finding is consistent with prior studies that showed higher rates of AF with accumulated training burden [5,7,17]. This association is independent of chronologic age and other known risk factors for AF, thus years of accumulated training appear to be more than simply a “marker” of increased chronologic age and accumulation of risk factors associated with AF. 

The lack of an association of training intensity and volume with reported AF in our cohort should be interpreted cautiously when applying this finding to a population of less avid runners. Our sample could be considered as composed of high-level runners—over half of our cohort reported completing a marathon and two-thirds reported they regularly incorporated interval or speed work into their training. Therefore, the possibility of missing an association between a high intensity training variable and AF in our cohort is a concern since the majority of our respondents were relatively high-level runners when compared to the running community in general. Additionally, the self-reported nature of our data is an obvious shortcoming, along with the lack of clinical verification of the underlying diagnosis of AF and the small sample size. It is certainly possible, for example, that a long-time runner who develops AF may be biased in his recall of his training duration or habits given the emerging data regarding this association. Of note, assessments of the use of self-reported lifetime exercise volumes have suggested that they have low validity and reliability [18]. The exclusion of other types of exercise routines also limits the generalizability. Lastly, the lack of additional clinical measures such as atrial size/dimensions, which is a known risk for AF in endurance athletes, limits the scope of this project.

Nonetheless, this study adds further insight into the growing body of data supporting a relationship between high intensity endurance exercise and the development of AF in aging runners. Specifically, it suggests that accumulated training years, but not specific training activities, promote the development of AF. Further studies prospectively examining specific modes of endurance exercise and the development of AF are needed to confirm these findings and further define how healthcare providers can most effectively counsel runners and endurance athletes on the role of lifetime training on the development of AF.

## Figures and Tables

**Figure 1 sports-07-00179-f001:**
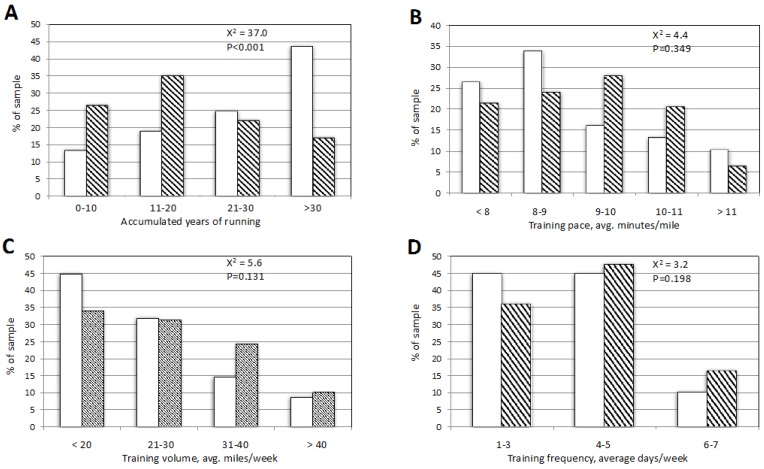
Distribution of runners reporting (solid bars) vs. those not reporting (hatched bars) atrial fibrillation stratified by (**A**) years of accumulated running; (**B**) training pace; (**C**) average miles/week; (**D**) average running/week.

**Table 1 sports-07-00179-t001:** Multivariable logistic regression, relationship of demographic, health and training characteristics with reported atrial fibrillation.

	Odds Ratio	95% CI	*p* Value
Chronologic Age (per year)	1.077	1.045–1.109	<0.0001
Hypertension	2.116	1.137–3.395	0.018
Years Running (per 10 year increment)	1.455	1.061–1.995	0.020
Diabetes Mellitus	2.081	0.405–10.688	0.380
Male Sex	1.372	0.618–3.047	0.437

In this model, chronologic age (Odds Ratio (OR) 1.077/year, 95% CI 1.045–1.109, *p* < 0.001), hypertension (OR 2.116, 95% CI 1.137–3.935, *p* = 0.018), and years of accumulated training (OR 1.455 per decade of training, 95% CI 1.061–1.995, *p* = 0.020) were independent predictors of AF. In contrast, diabetes mellitus and male sex were not independent predictors of the development of AF.

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
