# Peer review of "An Assessment of Training Characteristics Associated with Atrial Fibrillation in Masters Runners"

_sports, 2019, doi:10.3390/sports7070179_

Round 1

Reviewer 1 Report

 The manuscript talks about an interesting aspect of the sports medicine especially for  the population aged > 35 yrs.

 The responsibility  of  the time of   exposition  and the  intensity  of the training  in the origin of the  AF  , has been largely discussed .

To improve  the  massage the paper could be little  modified  including more details  regarding the kind  of the   training practiced . : aerobic  or mixed  exercise . ?

The  authors refer the data  to  the intensity  of exercise , however  the kind of exercise  described  includes only the runners . Have you some data regarding  the  eventual association in case  of different king of sports practiced ?Can  you support  or exclude  this  hypothesis in the discussion session? . In any case  the missing  of this data need to be explicated as a limit of the manuscript .  

In addition can  you add some  information of the myocardial chambers  dimension especially  for the left atria volume  and mass . We know  that in runners the kind of the  training is often determinant  to induce an enlargement  of the atria chambers due to  the continuous overloading .

 This alone can support  the AF onset and often  the  recurrence . This aspect needs to  be cited  in the  text

Author Response

Reviewer 1

Comment 1: To improve the message, the paper could be little modified including more details regarding the kind of the training practiced: aerobic or mixed exercise?

Response 1: We agree that more information regarding the training exercises would be helpful but unfortunately, but as our study was focusing on runners we only gathered the following details regarding running-specific training: duration (years of participation), volume (weekly mileage), intensity (average training pace), and frequency (average training days/week). We did gather information on interval training, but did not ask about cross training or non-running exercise.

Comment 2: The authors refer the data to the intensity of exercise, however the kind of exercise described includes only the runners. Have you some data regarding the eventual association in case of different king of sports practiced? Can you support or exclude this hypothesis in the discussion session? In any case, the missing of this data need to be explicated as a limit of the manuscript. 

Response 2: The hope is to conduct further research on different types of exercise to see if the results remain consistent. This survey was only focused on runner as respondents were recruited using advertisements in national running publication (Running Times, Rodale Press,

Emmaus, Pennsylvania) and on multiple running-related internet message boards. The limitation section now includes the following:

Discussion: “The exclusion of other types of exercise routines also limits the generalizability.”

Comment 3: In addition, can you add some information of the myocardial chambers dimension especially for the left atria volume and mass. We know that in runners the kind of the training is often determinant to induce an enlargement of the atria chambers due to the continuous overloading. This alone can support the AF onset and often the recurrence. This aspect needs to be cited in the text

Response 3: This is a great point so we updated our discussion to include that limitation as we do not have any clinical information regarding atrial size/dimensions.

Discussion: “Lastly, the lack of additional clinical measures such as atrial size/dimensions which is a known risk for AF in endurance athletes limits the scope of this project.”

Reviewer 2 Report

In this survey study, Matsumura and Abbatemarco investigated the potential link between long-term endurance exercise and the development of atrial fibrillation (AF). The authors identified by using a multivariate analysis that age, hypertension and accumulated years of running were independent predictors of the development of AF. This study provides some interesting results. Please consider the following comments.     

1. The study period is not clear. It is mentioned that the survey started in July 2014 but it is not mentioned when the survey ended.

2. Please provide the inclusion and exclusion criteria of the survey.

3. Why did you only focus on runners over 35 years old? Please justify this threshold value.

4. Please further detail your sample size. How many runners have completed the survey? How many runners did you exclude from the analysis and why? It would be informative for readers to provide a flowchart.

5. How could you be sure that the cardiac arrhythmia reported by the respondents was actually AF? It is a crucial point for your analysis, which must be clearly detailed in the methods section.

6. The statistical analysis needs to be further detailed:

-  What test did you use to compare quantitative variables between runners with and without AF?     

- Regarding the multivariate analysis, what p value in univariate analysis did you consider as significant to include variables in the multivariate analysis? In addition, what model of multivariate analysis did you use (enter, stepwise, forward, backward…)?

7. The baseline characteristics of your cohort, especially information regarding the training practices, should be further detailed (you only reported the mean age and the proportion of male).  

8. Please provide in a table all the data of interest you collected during the survey for runners with and without AF and the results of the statistical comparison between the two groups. It is of importance to clearly understand which variables you did not consider for the multivariate analysis despite statistical significance in univariate analysis.   

9. In the results, lines 68-69: “A current or prior diagnosis of AF was reported by 69 (2.4%)”. Does it mean that only 69 runners in this survey experienced AF? Please clarify this point.

10. Regarding the Figure 1, it is not clear whether the x² and the p value provided for each panel concern the “whole panel” or only some specific categories within each panel. Please clarify this point and indicate on the figure the statistical significance when necessary.

11. In the discussion, lines 99-100: “The present study is the first to focus on older runners (…)”. I am not convinced that runners over 35 years old can really be considered as older runners. This sentence should be rewritten.

12. In the discussion, it could be interesting to briefly describe your assumptions to explain your results. Please also discuss the limitations of your study (the small sample size of runners self-reporting AF, the potential lack of certainty regarding the diagnosis of AF…).

Author Response

Reviewer #2

Comment 1: The study period is not clear. It is mentioned that the survey started in July 2014 but it is not mentioned when the survey ended.[JA1] 

Response 1: The study collected data for a 2 month period in July and August 2013 (not 2014, this was an error)

Comment 2: Please provide the inclusion and exclusion criteria of the survey.

Response 2: The survey was open for anyone who responded to the advertisements online and in Running Times magazine.  For the present study we included all runners who answered the question regarding whether they had ever been diagnosed with atrial fibrillation. We included the following to the methods:

“Any runner who responded to the survey and was 18 years of age or older was invited to participate. For the present study all participants who responded to the question regarding history of atrial fibrillation were included.”

Comment 3: Why did you only focus on runners over 35 years old? Please justify this threshold value.

Response 3: This is a good question- we chose 35 years of age as a lower age limit because this is the age at which runners are initially classified as “submasters.”  In our analysis of AF we did examine the prevalence of a report of AF by age, and as expected runners were found to be more likely to report AF with increasing age.

To your point we proposed changing our manuscript title to An Assessment of Training Characteristics Associated with Atrial Fibrillation in Masters Runners”  to better distinguish the cohort as a class of athletes rather than “older” in a pure chronologic sense

Comment 4: Please further detail your sample size. How many runners have completed the survey? How many runners did you exclude from the analysis and why? It would be informative for readers to provide a flowchart.[JA2] 

Response 4: This information was added below (in text rather than flowsheet):

“For the present study survey responses were received from 3517 participants. Of the 3517 respondents 2,819 (80.1%) provided complete answers regarding history of AF and  training history andwere included in the assessment”

Comment 5: How could you be sure that the cardiac arrhythmia reported by the respondents was actually AF? It is a crucial point for your analysis, which must be clearly detailed in the methods section[JA3] .

Response 5: We agree that this is a crucial part of the analysis and our AF sample was based only on self-report of this arrhythmia.  This is an obvious weakness of the study as AF was not confirmed by ECG or rhythm monitoring. 

Comment 6: The statistical analysis needs to be further detailed:

What test did you use to compare quantitative variables between runners with and without AF?    

- Regarding the multivariate analysis, what p value in univariate analysis did you consider as significant to include variables in the multivariate analysis? In addition, what model of multivariate analysis did you use (enter, stepwise, forward, backward…)?

Response 6: Chi squared analysis was used to compare quantitative variables between runners with an without AF.  A p-value <0.05 on univariate analysis was used to decide variables to include in our multivariate analysis.

Comment 7: The baseline characteristics of your cohort, especially information regarding the training practices, should be further detailed (you only reported the mean age and the proportion of male).  [JA4] 

Response 7:  We added demographic/disease details to the text results:

“The mean age of the respondents was 49.1 years (range 35-90) and 68.2% were male.  Disease history was as follows: hypertension 20.9%, hyperlipidemia 21.9%, diabetes mellitus 1.4%, famly history of cardiovascular disease 40.7%. “

Comment 8: Please provide in a table all the data of interest you collected during the survey for runners with and without AF and the results of the statistical comparison between the two groups. It is of importance to clearly understand which variables you did not consider for the multivariate analysis despite statistical significance in univariate analysis.  

Response 8:   Variables we examined fall into 2 categories:  Training specific variables (ie figure 1 the manuscript) and factors that have been shown to be associated with the development of AF, specificially: age, male sex, diabetes, and hypertension (references have been added to the manuscript to support these variables).  Some variables known to be associated with AF (BMI, alcohol use) were not surveyed and could not be reported- of these, it would be expected that BMI would not have a significant influence over AF in this cohort given they are endurance athletes and likely for the most part have low to normal BMIs. In addition, while we had data on smoking, we did not have robust data on duration of smoking (we only assessed “smoking of 100+ cigarettes or more as + smoking history) we did not include this in the univariate analysis.  To support this decision, the Rotterdam study found no association of a status of  “former smoker” with AF, as opposed to a positive association of “current smoker” with AF (Heering J, et al, Am Heart J, 2008).

The following text was added/modified in the methods section, with new reference citations added:

“Additionally, they were asked about specific characteristics known to be associated with AF factors (age, sex, diabetes, hypertension). Some traditional risk factors were not assessed in the survey and thus were not able to be measured in this study (obesity, alcohol use, smoking)[14-15].”

It should be noted that univariate analysis of risk factors and association with AF is presented in the results (if the reviewers feel a table should also be included we can do this):

“Significant differences were found in mean age (59.8 vs. 48.4 years, p<0.001), male sex (85.5% vs. 67.8%, p=0.003), and report of hypertension (47.8% vs. 20.2%, p<0.001) and diabetes mellitus (7.2% vs. 1.3%, p<0.001) between those with vs. without AF, respectively.  “

Comment 9: In the results, lines 68-69: “A current or prior diagnosis of AF was reported by 69 (2.4%)”. Does it mean that only 69 runners in this survey experienced AF? Please clarify this point.

Response 9: Yes only 2.4% of the study population had atrial fibrillation. We have updated our limitation section to include the following:

Discussion: “Additionally, the self-reported nature of our data is an obvious shortcoming along with the lack of clinical verification of the underlying diagnosis of AF and the small sample size. It is certainly possible, for example, that a long-time runner who develops AF may be biased in his recall of his training duration or habits given the emerging data regarding this association. The exclusion of other types of exercise routines also limits the generalizability. Lastly, the lack of additional clinical measures such as atrial size/dimensions which is a known risk for AF in endurance athletes limits the scope of this project.”

Comment 10: Regarding the Figure 1, it is not clear whether the x² and the p value provided for each panel concern the “whole panel” or only some specific categories within each panel. Please clarify this point and indicate on the figure the statistical significance when necessary.

Response 10: The x2  and p-values concerned trends for each panel (comparison) as a whole

Comment 11: In the discussion, lines 99-100: “The present study is the first to focus on older runners (…)”. I am not convinced that runners over 35 years old can really be considered as older runners. This sentence should be rewritten.

Response 11: The sentence has been rewritten as:

Discussion: “The present study is the first to focus on runners age 35 and above while examining the relationship of specific lifelong training regimens and variability in running volume and intensity with AF.”

Comment 12: In the discussion, it could be interesting to briefly describe your assumptions to explain your results. Please also discuss the limitations of your study (the small sample size of runners self-reporting AF, the potential lack of certainty regarding the diagnosis of AF…).

Response 12: This is an excellent point and was also raised by the other reviewer. That section now reads

Discussion: “Additionally, the self-reported nature of our data is an obvious shortcoming along with the lack of clinical verification of the underlying diagnosis of AF and the small sample size. It is certainly possible, for example, that a long-time runner who develops AF may be biased in his recall of his training duration or habits given the emerging data regarding this association. The exclusion of other types of exercise routines also limits the generalizability. Lastly, the lack of additional clinical measures such as atrial size/dimensions which is a known risk for AF in endurance athletes limits the scope of this project.”

Reviewer 3 Report

This is an interesting study, but with a relatively small sample size of those with AF (not unexpected).

Abstract should note the number of participants reporting AF, lest the total sample size potentially mislead the reader.

Methods

Line 55: Provide reference for parameters mentioned being traditional risk factors.

Results

Figure 1 and Table 1 are formatted incorrectly and thus are unintelligible.

Discussion

Line 99: State that it is a relatively rare side effect and cite the prevalence from previous research e.g. in the general population.

Discussion should acknowledge the relatively limited sample size of those with AF (since it is a relatively rare condition) in terms of the power of the statistical analysis and the overall application of the findings to the wider population.

Author Response

Reviewer #3

Comment 1: Abstract should note the number of participants reporting AF, lest the total sample size potentially mislead the reader.

Response 1: This is an excellent point. That section was updated to include the following:

Abstract: “A total 2,819 runners participated and 69 (2.4%) reported a current or prior diagnosis of AF.”

Comment 2: Line 55: Provide reference for parameters mentioned being traditional risk factors.

Response 2: References for “traditional” AF risk factors were added to the reference section. While this list is certainly not exhaustive, the point of the study was to examine whether long-term participation in endurance exercise is an independent risk factor beyond traditional/common factors known to be associated with the development of AF.

Comment 3: Figure 1 and Table 1 are formatted incorrectly and thus are unintelligible.

Response 3: We will make sure the items are uploaded correctly but the other reviewers were able to view them correctly and make appropriate comments.

Comment 4: Line 99: State that it is a relatively rare side effect and cite the prevalence from previous research e.g. in the general population.

Response 4: This is an insightful comment and we have now updated that sentence to include:

Discussion: “While there is little downside to long term, high level endurance exercise, the development of AF has emerged as a rare “side effect” of this endeavor with studies suggesting a rate between 6-10% [1-3].”

Comment 5: Discussion should acknowledge the relatively limited sample size of those with AF (since it is a relatively rare condition) in terms of the power of the statistical analysis and the overall application of the findings to the wider population.

Response 5: This is an excellent point and was also raised by the other reviewer. That section now reads:

Discussion: “Additionally, the self-reported nature of our data is an obvious shortcoming along with the lack of clinical verification of the underlying diagnosis of AF and the small sample size. It is certainly possible, for example, that a long-time runner who develops AF may be biased in his recall of his training duration or habits given the emerging data regarding this association. The exclusion of other types of exercise routines also limits the generalizability. Lastly, the lack of additional clinical measures such as atrial size/dimensions which is a known risk for AF in endurance athletes limits the scope of this project.”